# Multi-resolution Spectral Coherence for Graph Generation with Score-based Diffusion

**Hyuna Cho    Minjae Jeong    Sooyeon Jeon    Sungsoo Ahn    Won Hwa Kim**
POSTECH, South Korea
{hyunacho, minjaetidtid, jsuyeon, sungsoo.ahn, wonhwa}@postech.ac.kr

## Abstract

Successful graph generation depends on the accurate estimation of the joint distribution of graph components such as nodes and edges from training data. While recent deep neural networks have demonstrated sampling of realistic graphs together with diffusion models, however, they still suffer from oversmoothing problems which are inherited from conventional graph convolution and thus high-frequency characteristics of nodes and edges become intractable. To overcome such issues and generate graphs with high fidelity, this paper introduces a novel approach that captures the dependency between nodes and edges at multiple resolutions in the spectral space. By modeling the joint distribution of node and edge signals in a shared graph wavelet space, together with a score-based diffusion model, we propose a Wavelet Graph Diffusion Model (Wave-GD) which lets us sample synthetic graphs with real-like frequency characteristics of nodes and edges. Experimental results on four representative benchmark datasets validate the superiority of the Wave-GD over existing approaches, highlighting its potential for a wide range of applications that involve graph data.

## 1   Introduction

The flexible representation of graphs lets us investigate a variety of real-world phenomena such as social networks [8,39], traffic flows [43,45], molecule structure [18,19,34,44], and brain networks [2, 22,28]. The rich expressive power of graphs inherits from embodying various combinations of nodes and edges within an irregular structure. However, the analysis of graph data is challenging due to the arbitrary and heterogeneous structure, as it becomes difficult to characterize robust patterns both within a graph and across different graphs for a downstream machine learning task.

The same issue arises in graph generation, where the objective is to learn the underlying graph distribution from training samples and thereby synthetic graphs with similar characteristics to the training data can be generated, i.e., sampled. For this, accurately estimating the joint distribution of the measurements from node features and graph structures is essential. Recent graph generative methods [13,35,37] have been successful in sampling realistic node and edge features, however, their generated samples often fail to embed the true relationships especially given in sparse connections between nodes. This problem often occurs as existing methods derive separate embeddings of nodes and edges considering them as different entities [37]. Moreover, successive layers of graph convolution [13,35] aggravate this issue by overly averaging signals on graphs, such that local details (i.e., high-frequency characteristics) become intractable.

**Key ideas.** Notice that the edge signals, e.g., edge weights and connectivity, should be coherent to node signals as the edges should properly explain the relationships between the nodes. Based on this homophily assumption, to address the aforementioned issues, we propose a novel generative model that precisely captures the dependency between nodes and edges in *multiple resolutions* on local-to-global structures. This is realized via spectral graph wavelet transform (SGWT) [9], which

37th Conference on Neural Information Processing Systems (NeurIPS 2023).

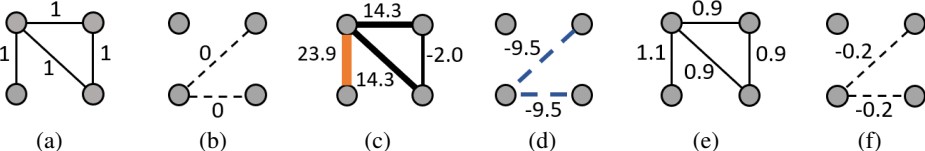

Figure 1: An example of SGWT with diverse scales that impacts the graph representation. (a) An undirected graph with an adjacency matrix $\mathbf{A}$, (b) Disconnectivity between nodes with dashed lines for $\mathbf{A}$. (c) Result from graph transform of $\mathbf{A}$ with a band-pass kernel $k(s) = sxe^{-sx}$ at scale $s = 1$ and (d) corresponding disconnectivity to (c). (e) Result from graph transform of $\mathbf{A}$ using the same band-pass kernel at $s = 0.1$. (f) Corresponding disconnectivity for (e). The edge thickness denotes the magnitude of connectivity, which is controlled by a scale parameter. Compared to $\mathbf{A}$, the sparse connection (orange) is significantly strengthened via SGWT at $s = 1$, and disconnectivities (blue) are also emphasized at the same scale.

enables the filtering of graph signals at a specific level of granularity. We transform the node and edge signals into a common graph wavelet space and obtain their underlying cross-characteristics, i.e., spectral coherence, via their inner product at different scales. We expect that sparse connections between nodes (i.e., higher variation) can be captured with high fidelity as high-frequency components in the spectral space. Fig. 1 shows an example of such an effect with wavelet transforms on edges, which accentuates certain connectivity and disconnectivity over a template graph. These accentuated (dis-)connectivities at specific scales can be characterized by a generative model with multiresolution wavelet filtering. Also, we further show that the scale-specific node and edge features computed in the spectral space can be easily and jointly obtained in a spatial graph convolutional form by applying wavelet filtering to either nodes or edges for efficiency.

**Proposed Framework.** We leverage this spectral coherence to model the joint distribution of graph node and edge features embedded in a common graph wavelet space with a score-based diffusion model [36]. As opposed to conventional approaches [13] where the scores for the joint distribution of nodes and edges are modeled with graph multi-head attention (GMH) [1] involving graph convolutions of the raw node and edge signals, our model calculates the scores over multiple scales of nodes and edges to capture different frequency characteristics. With diffusion and reverse denoising processes in multi-resolution, our Wavelet Graph Diffusion Model (Wave-GD) accurately estimates the joint distribution of the node and edge features preserving their graph frequency characteristics.

**Contributions.** To this end, we summarize our main contributions as follows: **1)** To obtain representative graph characteristics of different fidelity, we propose a novel concept to capture spectral coherence between nodes and edges in multiple scales in a common graph wavelet space. **2)** Furthermore, we show that spectral coherence can be obtained in the graph domain as a spatial graph convolutional formulation, which allows us to perform the SGWT only on the edge signals for efficiency. **3)** We propose a training scheme on the multi-resolution at which the optimal joint frequency characteristics of graph nodes and edges can be captured from training samples. The Wave-GD was tested on four representative benchmarks for graph generation which demonstrated outperforming results over state-of-the-art baselines. Extensive experiments on real-world, synthetic, and biochemical graph datasets validate the generality of the method, suggesting its potential for diverse applications with graph samples.

## 2 Related Work

Deep graph generative models aim to learn the underlying distribution of node and edge signals from the training set. They can be categorized into (1) autoregressive models that construct graphs by the iterative generation of a node or an edge signal at each step or (2) one-shot models that simultaneously update all the signals at once.

**Autoregressive models.** Autoregressive models are widely used for graph generation since they are based on a straightforward way to iteratively update the graph structure conditioned on an incomplete graph generated so far. Prior works have parameterized the autoregressive models using recurrent neural network (RNN) [29, 41, 42], variational autoencoder (VAE) [11, 12], and normalizing flow [20, 34]. However, they suffer from high complexity since the required number of updates to generate the graph grows with the size of the graph. Furthermore, their generation is sensitive to the order of nodes being generated, i.e., autoregressive models may assign different probabilities for the construction of the same graph executed in a different way.

**One-shot models.** One-shot models alleviate the issues of autoregressive models by simultaneous generation of node and edge signals. This potentially reduces the complexity of generation since the number of generation steps no longer depends on the graph size. Furthermore, their generation is often permutation invariant and assigns the same probability for generating a graph regardless of the permutation. Researchers have considered parameterizing these models using generative adversarial network (GAN) [4], VAE [21], normalizing flow models [23, 44], and diffusion models [13, 25, 37].

**Diffusion models.** In particular, diffusion models are recently catching attention due to their extraordinary capability in learning the joint distributions of graph components such as nodes and edges. They are based on (1) defining a forward diffusion process to sequentially corrupt a graph using noise distributions and (2) training a reverse diffusion process parameterized by a graph neural network to reconstruct the original graph. Existing works mainly differ by defining the diffusion processes in the continuous space [13] or the discrete space [10, 25, 37].

## 3 Generating Graphs with Robust Spectral Characteristics

### 3.1 Preliminary: Graph Wavelet Transform

Let $G = (\mathbf{X}, \mathbf{A})$ be an undirected graph, where $\mathbf{X} \in \mathbb{R}^{N \times F}$ represents $F$-dimensional node features for $N$ nodes and $\mathbf{A} \in \mathbb{R}^{N \times N}$ is an adjacency matrix representing connectivity among the nodes. The graph Laplacian is defined as $\mathbf{L} = \mathbf{D} - \mathbf{A}$, where $\mathbf{D}$ is a diagonal degree matrix. As the $\mathbf{L}$ is real and positive semi-definite, it has a set of orthonormal eigenvectors $\mathbf{U} = [u_1, u_2, ..., u_N]$ and corresponding non-negative real eigenvalues $\mathbf{\Lambda} = diag(\lambda_1, ..., \lambda_N)$. Decomposing this Laplacian as $\mathbf{L} = \mathbf{U}\mathbf{\Lambda}\mathbf{U}^T$, the connectivity, complexity, and spectral properties of the graph are characterized. Specifically, its eigenvectors and eigenvalues identify the frequencies and mode of vibrations of a graph. Eigenvectors with small eigenvalues represent large connected components in a graph structure and slow-varying signals among nodes. On the other hand, eigenvectors associated with large eigenvalues indicate the sparse and disconnected signals in the graph [22, 26, 40].

To capture localized characteristics of signals on arbitrary graph structures, the spectral graph wavelet transform [9] is defined by constructing a set of graph wavelet bases $\psi_s = (\psi_{s_1}, \psi_{s_2}, ..., \psi_{s_J})$ with $J$ scales where $\psi_s = \mathbf{U}k(s\mathbf{\Lambda})\mathbf{U}^T$. The $\psi_s$ is a realization of a kernel function $k(\cdot)$ in the spectral domain localized with $\delta_n$ in the graph space, which captures the local characteristics of the graph at each node. The scale $s$ selects a specific bandwidth in the spectral space which corresponds to the range of locality in the original graph space. The choice of wavelet basis, i.e., mother wavelet, depends on the shape of $k(\cdot)$, which may vary depending on a target task such as smoothing or band-pass filtering. In this work, we used both types of kernel functions: (1) a band-pass kernel $k(s) = sxe^{-sx}$ to capture both low and high frequency of graphs and (2) a low-pass kernel $k(s) = e^{-sx}$ to capture cluster-like and smoothed features from graphs.

Using the bases $\psi_s$, graph wavelet transform projects a signal $x$ (e.g., node feature or connectivity) from the graph domain into the spectral domain as

$$W_x(s) = < \psi_s, x > = \psi_s x, \tag{1}$$

which yields a wavelet coefficient $W_x(s)$. This wavelet coefficient is a scale-filtered signal defined in the wavelet space. When admissibility condition on $k(\cdot)$ is satisfied [24], the inverse transform completely reconstructs the signal $x$ by projecting $W_x(s)$ back to the graph domain as follows:

$$x = \frac{1}{C_k} \int_0^\infty \psi_s \cdot W_x(s) \frac{ds}{s} \tag{2}$$

with an admissibility constant $C_k = \int_0^\infty \frac{k(x)^2}{x} dx < \infty$.

### 3.2 Adaptive Multi-Resolution Representation of Nodes and Edges

To capture both fine and coarse graph features from an irregular graph structure, the filtering operation is performed over multiple scales. The multi-resolution representations of graph nodes and edges are derived by decomposing the graph signals $\mathbf{X}$ and $\mathbf{A}$ with scales $\{s_i\}_{i=1}^J$ using (1), which result in $\{W_\mathbf{X}(s_i)\}_{i=1}^J$ and $\{W_\mathbf{A}(s_i)\}_{i=1}^J$ in the *spectral domain*. These are wavelet coefficients that contain different levels of local and global details of graph node and edge signals in a multi-resolution fashion.

To obtain the multi-resolution representations of these graph signals in the *graph space*, the inverse transform in the Eq. 2 is performed on the filtered coefficients using the bases $\psi_s$ at the same scales.

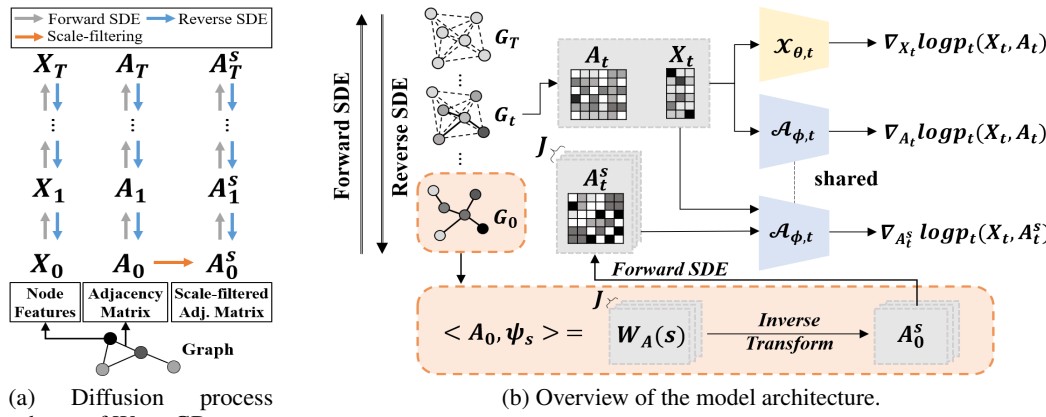

(a) Diffusion process scheme of Wave-GD.

(b) Overview of the model architecture.

Figure 2: (a) Schematic diagram of Wave-GD with multi-resolution diffusion. Note that the scale-filtering is applied to the $\mathbf{A}_0$, not to the $\mathbf{A}_t$. (b) Overview of Wave-GD that jointly estimates partial scores of node features $\mathbf{X}$ and edges $\mathbf{A}$ along with scale-filtered edges $\mathbf{A}^s$. We first perform multi-resolution filtering on the given edges $\mathbf{A}_0$ at $t = 0$ using graph wavelet basis $\psi_s$ with trainable scale $s$. Forward diffusion processes smoothly transform $\mathbf{X}$, $\mathbf{A}$, and $\mathbf{A}^s$ into noises in parallel and they are reconstructed by solving reverse-time SDEs. During the reconstruction, the knowledge across multi-resolution is shared in the score-based model $\mathcal{A}_{\phi,t}$. Also, the information in separate edge and node components is effectively entangled by obtaining their spectral coherence at various resolutions.

As the Eq. 2 is the superposition of multi-resolution representation of $x$ over scales at $s \in [0, \infty)$, it allows us to define a scale-filtered signal in the graph space at specific $s$ as

$$x^s = \psi_s \cdot W_x(s) = \mathbf{U}k^2(s\Lambda)\mathbf{U}^T x, \tag{3}$$

where $k^2(s\Lambda) = diag(k^2(s\lambda_1), ..., k^2(s\lambda_N))$ is a diagonal matrix [14, 22]. Therefore, the $\mathbf{X}$ and $\mathbf{A}$ at multiple resolutions in the spatial domain are defined as $\{\mathbf{X}^{s_i}\}_{i=1}^J = \{\mathbf{U}k^2(s_i\Lambda)\mathbf{U}^T\mathbf{X}\}_{i=1}^J$ and $\{\mathbf{A}^{s_i}\}_{i=1}^J = \{\mathbf{U}k^2(s_i\Lambda)\mathbf{U}^T\mathbf{A}\}_{i=1}^J$, respectively. Within our proposed framework, we make these scales *trainable* such that the local-to-global characteristics can be obtained adaptively.

### 3.3 Refining Spectral Coherence between Nodes and Edges in Multi-resolution Fashion

As the graph features $\mathbf{X}$ and $\mathbf{A}$ should be coherent, i.e., connectivities should explain the relationship among features from different nodes, we hypothesize that they should share similarities even when observed from different resolutions. To capture the key components that are shared on both edges and nodes at multi-resolution, we measure the underlying cross-characteristics between nodes and edges in each scale in the spectral space. We quantify their coherence as a dot product as $W_\mathbf{A}(s) \cdot W_\mathbf{X}(s)$ and use this spectral coherence as a feature that represents a graph at a specific resolution $s$, which is used to estimate the score function in multiple resolutions later in Section 3.5.

**Lemma 1.** *Let $G$ be an undirected graph, with a node signal $\mathbf{X}$ and a symmetric adjacency matrix $\mathbf{A}$. Given wavelet coefficients $W_\mathbf{A}(s)$ and $W_\mathbf{X}(s)$, which are spectral representations of $\mathbf{X}$ and $\mathbf{A}$ at a specific scale $s$, a dot product of these wavelet coefficients is equivalent to the product between the scale-filtered edge signals and the node signals in the graph space, i.e., $W_\mathbf{A}(s) \cdot W_\mathbf{X}(s) = \mathbf{A}^s\mathbf{X}$.*

By Lemma 1, we can efficiently extract multi-resolution coherence by applying the filtering only on the edges. Using the $J$ number of $\mathbf{A}^s$ along with the given data $G = (\mathbf{A}, \mathbf{X})$, we introduce our simple yet effective graph generative framework through the stochastic differential equations (SDEs) in Sec. 3.4 and explain how Lemma 1 is used in the generative model in Sec. 3.5. The proof of Lemma 1 is given in Appendix A.

### 3.4 Wave-GD: Wavelet Graph Diffusion Model via Multi-resolution

Now we describe our graph generative model which captures the dependency of node and edge on both local and global structures with adaptive spectral filtering. The overall description of Wave-GD is given in Fig. 2, which demonstrates (a) multi-resolution diffusion strategy and (b) overall model architecture. In particular, we enhance the existing diffusion models for graphs [13] using the spectral coherence introduced in Lemma 1. The building blocks of our model are as follows: 1) the forward

diffusion to corrupt the graph distribution in a continuous domain, 2) the reverse diffusion to recover the original graph, and 3) the training objective for our model.

**Multi-resolution forward diffusion process.** Building upon the work of GDSS [13], we propose a new diffusion process that sequentially corrupts a graph structure at different levels of resolution based on the Itô SDE [32]. As the forward diffusion process is modeled by a SDE which describes a process of random variables, $\{\mathbf{X}_t, \{\mathbf{A}_t^{s_i}\}_{i=0}^J\}$ with $\mathbf{A}_t^{s_0} = \mathbf{A}_t$ at timestep $t \in [0, T]$ are diffused as follows:

$$d\mathbf{X}_t = \mathbf{f}_{\mathbf{X}}(\mathbf{X}_t, t)dt + \sigma_{\mathbf{X},t}d\mathbf{w}_{\mathbf{X}},$$
$$d\mathbf{A}_t^{s_i} = \mathbf{f}_{\mathbf{A}^{s_i}}(\mathbf{A}_t^{s_i}, t)dt + \sigma_{\mathbf{A}^{s_i},t}d\mathbf{w}_{\mathbf{A}^{s_i}}, \quad (4)$$

where $\mathbf{f}_{\mathbf{X}} : \mathbb{R}^{N \times F} \to \mathbb{R}^{N \times F}$ and $\mathbf{f}_{\mathbf{A}^{s_i}} : \mathbb{R}^{N \times N} \to \mathbb{R}^{N \times N}$ are drift functions, $\sigma_{\mathbf{X},t}$ and $\sigma_{\mathbf{A}^{s_i},t}$ are scalar diffusion coefficients, and $\mathbf{w}_{\mathbf{X}}$ and $\mathbf{w}_{\mathbf{A}^{s_i}}$ are standard Wiener processes for $\mathbf{X}_t$ and $\{\mathbf{A}_t^{s_i}\}_{i=0}^J$, respectively. As this diffusion process is simulated at varying levels of granularity in a multi-resolution scheme, it is likely that certain frequency characteristics are accentuated and the highlighted information will be preserved from corruption. In other words, at an arbitrary timestep $t$, the scale-filtered edges $\{\mathbf{A}_t^{s_i}\}_{i=1}^J$ are highly likely to contain richer information of edges for score estimation than the given $\mathbf{A}_t$ as their edge signals are adaptively controlled and highlighted by trainable scales.

**Learning the reverse diffusion process.** Let $G_t = (\mathbf{X}_t, \mathbf{A}_t)$ be a given graph at timestep $t$ and $\{G_t^{s_i} = (\mathbf{X}_t, \mathbf{A}_t^{s_i})\}_{i=0}^J$ be a set of graphs for multiple scales at the same timestep. Given the multi-resolution forward diffusion process, we generate graphs from solving the associated SDE backward in time. To be specific, the reverse diffusion process can be formulated as follows:

$$d\mathbf{X}_t = [\mathbf{f}_{\mathbf{X}}(\mathbf{X}_t, t) - \sigma_{\mathbf{X},t}^2 \nabla_{\mathbf{X}_t} \log p_t(G_t)]d\bar{t} + \sigma_{\mathbf{X},t}d\bar{\mathbf{w}}_{\mathbf{X}},$$
$$d\mathbf{A}_t^{s_i} = [\mathbf{f}_{\mathbf{A}^{s_i}}(\mathbf{A}_t^{s_i}, t) - \sigma_{\mathbf{A}^{s_i},t}^2 \nabla_{\mathbf{A}_t^{s_i}} \log p_t(G_t^{s_i})]d\bar{t} + \sigma_{\mathbf{A}^{s_i},t}d\bar{\mathbf{w}}_{\mathbf{A}^{s_i}}, \quad (5)$$

where $p_t(G_t)$ and $p_t(G_t^{s_i})$ denote the distribution of the forward diffusion process evaluated at time $t$. This reverse-time SDE recovers the original data distribution $p_0(\mathbf{X}_0, \{\mathbf{A}_0^{s_i}\}_{i=0}^J)$ with the standard reverse-time Wiener processes $\bar{\mathbf{w}}_{\mathbf{X}}$ and $\bar{\mathbf{w}}_{\mathbf{A}^{s_i}}$ in which the direction of time is reversed as $d\bar{t}$.

However, since the score functions of the joint distribution, i.e., $\nabla_{\mathbf{X}_t} \log p_t(G_t)$ and $\nabla_{\mathbf{A}_t^{s_i}} \log p_t(G_t^{s_i})$, are unknown, we train a neural network to estimate them using the denoising score matching loss [38]:

$$L_{\mathbf{X}} = \min_{\theta} \mathbb{E}_t \left[ \lambda_{\mathbf{X}}(t) \mathbb{E}_{G_0, G_t|G_0}[\|\mathcal{X}_{\theta,t}(G_t) - \nabla_{\mathbf{X}_t} \log p_{0t}(\mathbf{X}_t|\mathbf{X}_0)\|_2^2]\right], \quad (6)$$

$$L_{\mathbf{A}} = \min_{\phi, \mathbf{s}} \mathbb{E}_t \left[ \lambda_{\mathbf{A}}(t) \mathbb{E}_{G_0, G_t|G_0}[\|\mathcal{A}_{\phi,t}(G_t) - \nabla_{\mathbf{A}_t} \log p_{0t}(\mathbf{A}_t|\mathbf{A}_0)\|_2^2] \right.$$
$$\left. + \lambda_{\mathbf{A}^{s_i}}(t) \sum_{i=1}^J \mathbb{E}_{G_0^{s_i}, G_t^{s_i}|G_0^{s_i}}[\|\mathcal{A}_{\phi,t}(G_t^{s_i}) - \nabla_{\mathbf{A}_t^{s_i}} \log p_{0t}(\mathbf{A}_t^{s_i}|\mathbf{A}_0^{s_i})\|_2^2]\right], \quad (7)$$

where $p_{0t}(\cdot)$ denotes the transition distribution from $p_0$ to $p_t$, $\mathcal{X}_{\theta,t}, \mathcal{A}_{\phi,t}$ are score-based models which are graph neural networks trained to approximate the true score functions. The $\lambda_{\mathbf{X}}(t), \lambda_{\mathbf{A}}(t), \lambda_{\mathbf{A}^{s_i}}(t)$ are time-varying scaling factors for individual loss terms and are defined as in [36].

Both our method and GDSS perform denoising score-matching to the partial scores of $X_t$ and $A_t$. However, our method additionally models the joint probability space of $X_t$ and $A_t^{s_i}$ via SDEs. This is realized by the loss in Eq. 7, and this operation allows a model to flexibly estimate the complex dependency between nodes and edges with multi-resolution SGWT. Also, note that the scales of SWGT $\{s_i\}_{i=1}^J$ in Eq. 7 are trainable so that multi-level granularities that characterize a graph distribution can be adaptively captured during training.

**Remark.** While we solve the reverse SDEs for $\{\mathbf{X}_t, \{\mathbf{A}_t^{s_i}\}_{i=0}^J\}$ in the *training*, we only need to solve the reverse diffusion processes (5) only for the node signal $\mathbf{X}$ and the non-filtered graph $\mathbf{A} = \mathbf{A}^{s_0}$ in the *sampling* phase. In other words, we do not generate $\{\mathbf{A}^{s_i}\}_{i=1}^J$ in the *sampling* phase. Also note that the parameters of the networks $\mathcal{A}_{\phi,t}$ are shared across different scales so that they can learn generalizable representations over different resolutions. As the shared parameters in the score function do not overfit to a specific scale, it also promotes stable training.

### 3.5 Score-based Joint Density Estimation with Multi-Resolution Coherence

Finally, we describe the score-based models $\mathcal{X}_{\theta,t}$ and $\mathcal{A}_{\phi,t}$ to better capture the spectral coherence over different resolutions. In particular, we revise the graph multi-head attention (GMH) [1] layer to better incorporate the spectral coherence between the edge and the node attributes. In the original GMH layer, node and edge attributes were encoded into query $Q$, key $K$, and value $V$ as follows:

$$Q = \mathbf{A}_t \mathbf{X}_t W_Q, \quad K = \mathbf{A}_t \mathbf{X}_t W_K, \quad V = \mathbf{A}_t \mathbf{X}_t W_V, \tag{8}$$

where $W_Q, W_K, W_V$ are trainable weights. This layer is repeated for each $\{s_i\}_{i=1}^{J}$ as follows:

$$Q^{s_i} = \mathbf{A}_t^{s_i} \mathbf{X}_t W_Q, \quad K^{s_i} = \mathbf{A}_t^{s_i} \mathbf{X}_t W_K, \quad V^{s_i} = \mathbf{A}_t^{s_i} \mathbf{X}_t W_V, \tag{9}$$

which incorporate spectral coherence at a specific scale $s_i$ as given in Lemma 1. Using $Q^{s_i}, K^{s_i}, V^{s_i}$ as the new query, key, and value for the GMH layer, one can incorporate the spectral coherence when training $W$'s. Consequently, the score-based models consist of multiple GMH layers followed by a multi-layer perceptron (MLP). We used the original GMH layer for the parameterization of $\mathcal{X}_{\theta,t}$, and the modified GMH layer is used for $\mathcal{A}_{\phi,t}$. Note that this parameterization preserves the permutation invariance of the original GMH layer, hence our score-based models are also permutation invariant.

## 4 Experiment

In this section, we quantitatively and qualitatively evaluate our method in comparison to various recent graph generative methods on four benchmark datasets. Ablation studies are also introduced to empirically analyze the roles of individual components within Wave-GD.

### 4.1 Datasets

We evaluated Wave-GD on four public datasets with varying sizes and characteristics, demonstrating its robustness and generalizability to generate high-quality graphs for diverse graph domains. **Ego-small** [33] consists of 200 small real sub-graphs from the Citeseer network dataset with $4 \leq N \leq 18$. **Community-small** consists of 100 randomly generated synthetic graphs with $12 \leq N \leq 20$. The graphs are constructed by two equal-sized communities and each community is generated by E-R model [5] and $0.05N$ inter-community edges are added with uniform probability as in previous works [17, 25]. **Grid** consists of randomly generated 100 standard 2D grid graphs with $100 \leq N \leq 400$. As all nodes are arranged in a regular lattice, the maximum number of edges per node is 4. **QM9** [31] is a molecular dataset with 133,885 molecules that are represented by attributed graphs with $1 \leq N \leq 9$, four node types, and three edge types.

### 4.2 Experimental Setup

For all datasets, we used 80% of the whole data as a train set and the rest 20% as a test set with the same split as in [13, 42]. As in [13], we sampled 10,000 molecules from the QM9 dataset, and for the remaining generic datasets, we sampled an equal number of data as the number of test data in each respective dataset. Also, the predictor-corrector sampler (PC-sampling) proposed in [36] was used to sample data with 1000 predictor and 1000 corrector steps.

We set two wavelet kernels $k(s)$: one is a low-pass filter $k(s) = e^{-sx}$ that captures signals in the low-frequency and the other $k(s) = sxe^{-sx}$ is a band-pass filter which is 0 at the origin. To obtain the combined effect of the low-pass and band-pass filters using a total of $J$ scales, edges at one scale are filtered using the low-pass filter. For the remaining $J-1$ scales, the band-pass filter is applied. This approach enables selective extraction and processing of different frequency components across the spectral domain. The number of scales $J$ was set to 11, 6, 4, and 6 for the Ego-small, Community-small, Grid, and QM9 datasets, respectively. More detailed experimental settings are given in Appendix B.

### 4.3 Experiment on Generic Graphs

To assess the versatility of Wave-GD, we used both real-world and synthetic datasets with varying sizes and characteristics. Experimental results on these diverse generic datasets demonstrate the robustness of Wave-GD in sampling graphs of various characteristics.

**Baselines and metrics.** For experiments on generic graph datasets (i.e., Ego-small, Community-small, and Grid), we compared the performance of our proposed method with the following autoregressive and one-shot graph generation methods: DeepGMG [16], GraphRNN [42], GraphAF [34],

Table 1: Generation results on generic graph datasets. MMD statistics between sampled graphs and test data are given, where smaller values indicate that the generated samples are more similar to the test set. The baseline results are from [13, 20, 25]. Mean from three replicates are reported, and their standard deviations are reported in Appendix C.

| | Method | Ego-small | | | | Community-small | | | | Grid | | | |
|---|---|---|---|---|---|---|---|---|---|---|---|---|---|
| | | Degree | Cluster | Orbit | Avg. | Degree | Cluster | Orbit | Avg. | Degree | Cluster | Orbit | Avg. |
| Autoreg. | DeepGMG [16] | 0.040 | 0.100 | 0.020 | 0.053 | 0.220 | 0.950 | 0.400 | 0.523 | - | - | - | - |
| | GraphRNN [42] | 0.090 | 0.220 | 0.003 | 0.104 | 0.080 | 0.120 | 0.040 | 0.080 | **0.064** | 0.043 | **0.021** | **0.043** |
| | GraphAF [34] | 0.030 | 0.110 | **0.001** | 0.047 | 0.180 | 0.200 | 0.020 | 0.133 | - | - | - | - |
| | GraphDF [20] | 0.040 | 0.130 | 0.010 | 0.060 | 0.060 | 0.120 | 0.030 | 0.070 | - | - | - | - |
| One-shot | GraphVAE [35] | 0.130 | 0.170 | 0.050 | 0.117 | 0.350 | 0.980 | 0.540 | 0.623 | 1.619 | **0.0** | 0.919 | 0.846 |
| | GNF [17] | 0.030 | 0.100 | **0.001** | 0.044 | 0.200 | 0.200 | 0.110 | 0.170 | - | - | - | - |
| | EDP-GNN [25] | 0.052 | 0.093 | 0.007 | 0.051 | 0.053 | 0.144 | 0.026 | 0.074 | 0.455 | 0.238 | 0.328 | 0.340 |
| | GDSS [13] | 0.021 | 0.024 | 0.007 | 0.017 | 0.045 | 0.086 | 0.007 | 0.046 | 0.111 | 0.005 | 0.070 | 0.062 |
| | DiGress [37]† | 0.017 | 0.021 | 0.010 | 0.016 | 0.028 | 0.115 | 0.009 | 0.050 | - | - | - | - |
| | Wave-GD (Ours) | **0.012** | **0.010** | 0.005 | **0.009** | **0.007** | 0.058 | **0.002** | **0.022** | 0.144 | 0.004 | **0.021** | 0.056 |

-: results not reported in the original manuscripts, †: results obtained with the official public code.

Table 2: Converged scales of generic graph datasets.

| Dataset | $J$ | low-pass | band-pass | | | | | | | | | |
|---|---|---|---|---|---|---|---|---|---|---|---|---|
| Ego-small | 11 | 45.4 | 46.8 | 46.6 | 43.7 | 40.6 | 38.1 | 37.0 | 31.3 | 29.0 | 25.6 | 20.0 |
| Community-small | 6 | 32.8 | 44.7 | 31.0 | 27.5 | 21.4 | 16.5 | - | - | - | - | - |
| Grid | 4 | 35.8 | 25.4 | 18.1 | 12.0 | - | - | - | - | - | - | - |

Table 3: Generation results on generic graph datasets using a larger number of samples. We compared 1024 generated samples with the given test set and evaluated their MMDs.

| Method | Ego-small | | | | Community-small | | | |
|---|---|---|---|---|---|---|---|---|
| | Degree | Cluster | Orbit | Avg. | Degree | Cluster | Orbit | Avg. |
| GraphRNN [42] | 0.040 | 0.050 | 0.060 | 0.050 | 0.030 | **0.010** | 0.010 | **0.017** |
| GNF [17] | 0.010 | 0.030 | **0.001** | 0.014 | 0.120 | 0.150 | 0.020 | 0.097 |
| EDP-GNN [25] | 0.010 | 0.025 | 0.003 | 0.013 | **0.006** | 0.127 | 0.018 | 0.050 |
| GDSS [13] | 0.023 | 0.020 | 0.005 | 0.016 | 0.029 | 0.068 | 0.004 | 0.034 |
| Wave-GD (Ours) | **0.010** | **0.018** | 0.005 | **0.011** | 0.016 | 0.077 | **0.001** | 0.031 |

GraphDF [20], GraphVAE [35], GNF [17], EDP-GNN [25], GDSS [13] and DiGress [37]. To evaluate the generation quality of the generated graphs, we used Maximum Mean Discrepancy (MMD) [7] to statistically measure similarities between probability distributions of sampled data and the real test data. Following prior works [13, 42], we compared distributions of three different graph statistics–degree, clustering coefficient, and orbit counts–to compute MMD between generated and test graphs.

**Main results.** Table 1 shows the comparisons of degree, clustering, orbit, and their averaged MMDs between the graphs generated from baselines and our method. The results show that our model significantly outperforms all baseline models for all metrics on Ego-small and Community-small datasets. Specifically, on the Ego-small dataset, Wave-GD demonstrated far smaller average MMDs by a margin of 0.108 over GraphVAE and at least 0.007 lower than DiGress. For the Community-small data, our method exhibited up to 0.601 lower MMD compared to GraphVAE and at least 0.024 better than GDSS. For the Grid dataset, our method outperformed all baselines in terms of orbit MMD and outperformed all one-shot baselines for the averaged MMD. Note that, while one-shot models struggle to generate large graphs due to their intricate and complex structures, our method achieved comparable results to GraphRNN on large grid graphs (with $100 \leq N \leq 400$). This highlights the effectiveness of our approach in learning graphs with varying sizes and structures through multi-resolution learning.

Table 2 shows converged scales for different datasets. Interestingly, the scales of the large Grid dataset (with $100 \leq N \leq 400$ nodes) were generally smaller compared to the Ego-small ($4 \leq N \leq 18$) and Community-small ($12 \leq N \leq 20$) datasets. Specifically, the converged scales of Ego-small ranged within $[20, 47]$ and the scales of Grid ranged within $[12, 36]$. Note that the smaller scales capture local graph features with higher frequencies. Therefore, the results demonstrate that capturing local and detailed graph representations is more critical when dealing with complex and large graphs, in

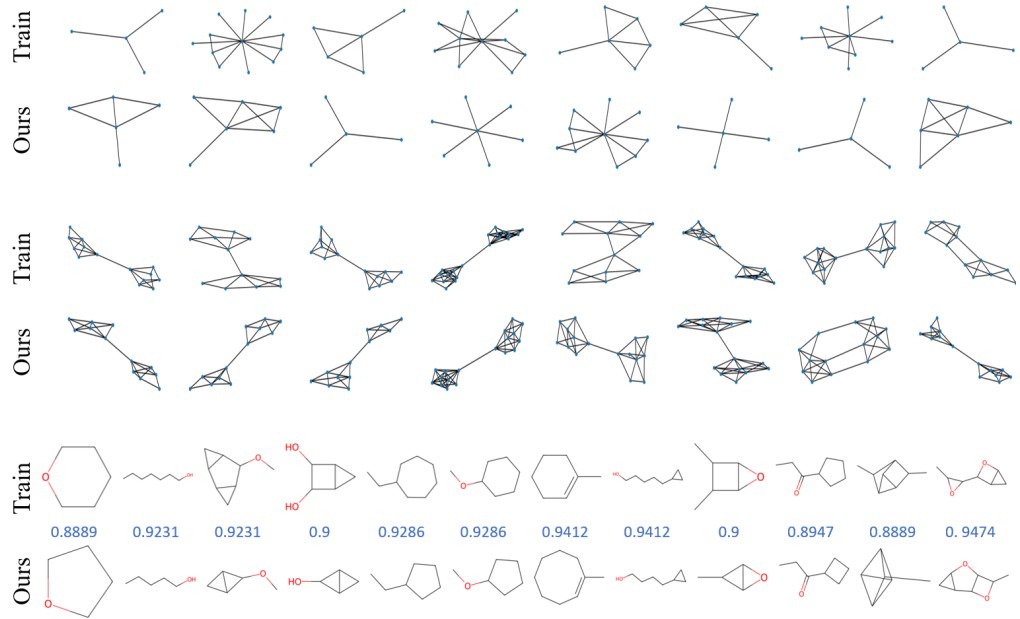

Figure 3: Visualization of training samples and generated graphs on Ego-small (top), Community-small (middle), and QM9 (bottom). The generated molecules from the QM9 dataset (bottom panel) are represented with the pairwise Tanimoto similarity based on the Morgan fingerprints [15], which show the one-to-one similarity to the real training samples at the same location in the upper panel.

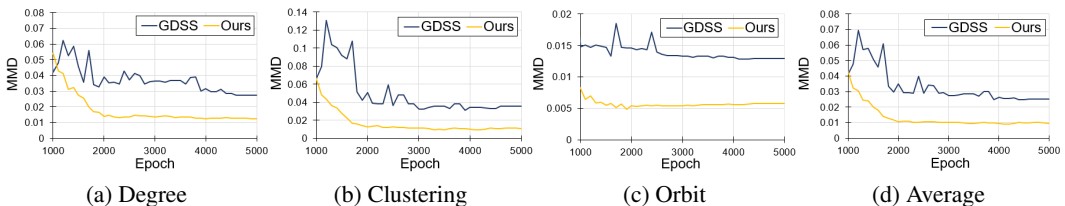

|     (a) Degree     |     (b) Clustering     |     (c) Orbit     |     (d) Average     |

Figure 4: Comparison of the MMDs between GDSS and Wave-GD on the Ego-small. The sample quality of Wave-GD is stable and converges faster for a) Degree, b) Clustering, c) Orbit, and d) their average.

comparison to smaller graphs. In other words, for the Ego-small and Community-small datasets, features in relatively low frequency (e.g., cluster-related features) were captured with larger scales.

**Experiments with larger sample size.** To provide a more definitive assessment of the generated sample quality, we conducted additional experiments by generating a larger number of samples compared to the test data, as in previous works [13, 17, 25]. For this, we sampled 1024 graphs for evaluating the MMD metrics for the Ego-small and Community-small datasets. As shown in Table 3, the lower MMDs of Wave-GD demonstrate superior performance to the baselines on Ego-small. For the Community-small dataset, GraphRNN performed better than ours on the averaged MMD, however, our method still surpasses recent score-based models [13, 25] and normalizing flow model [17] with at most 0.066 margin. Along with these quantitative results, we present the actual visualization of the generated samples from Wave-GD in Fig. 3, which are highly similar to the training samples.

**Stability analysis.** In addition to MMDs, we present a stability analysis of Wave-GD in Fig. 4. The figure presents a comparison of the averaged MMDs between GDSS and our method on the Ego-small dataset. Specifically, between 2000-5000 epochs, while the difference between the maximum and minimum values of the averaged MMD of GDSS is 0.016, Wave-GD shows better stability with a difference of ~0.0018 at the same epochs, indicating that our method can much more robustly generate high-quality samples.

### 4.4 Experiment on Molecular Graphs

In addition to the generic graphs, we evaluated Wave-GD on a molecular benchmark dataset, which contains complex chemical bonds between atoms (i.e., nodes) and edges. By evaluating Wave-GD on

Table 4: Generation results on QM9 molecular dataset. The baseline results are taken from [13, 20, 34] and all results are presented without validity correction. We report the mean of three runs and their standard deviations. The best performance is marked in bold and the second-best result is indicated by an underline.

| | Method | Validity (%)↑ | Novelty (%)↑ | Uniq. (%)↑ | Avg. (%)↑ | NSPDK↓ | FCD↓ | time (s)↓ |
|---|---|---|---|---|---|---|---|---|
| AR | GraphAF [34] | 67 | 88.83 | 94.51 | 83.45 | $0.020 \pm 3e^{-3}$ | $5.27 \pm .4$ | $2.52e^3$ |
| | GraphDF [20] | 82.67 | **98.10** | 97.62 | 92.80 | $0.063 \pm 1e^{-3}$ | $10.82 \pm .0$ | $5.35e^4$ |
| One-shot | MoFlow [44] | $91.36 \pm 1.2$ | $94.72 \pm .8$ | $98.65 \pm .6$ | $\underline{94.91}$ | $0.017 \pm 3e^{-3}$ | $4.47 \pm .6$ | $\mathbf{4.60e^0}$ |
| | EDP-GNN [25] | $47.52 \pm 3.6$ | $86.58 \pm 1.9$ | $\mathbf{99.25 \pm .1}$ | 77.78 | $0.005 \pm 1e^{-3}$ | $\underline{2.68 \pm .2}$ | $4.40e^3$ |
| | GraphEBM [18] | $8.22 \pm 2.2$ | $\underline{97.01 \pm .2}$ | $97.90 \pm .1$ | 67.71 | $0.030 \pm 4e^{-3}$ | $6.14 \pm .4$ | $\underline{3.71e^1}$ |
| | GDSS [13] | $95.72 \pm 1.9$ | $86.27 \pm 2.3$ | $98.46 \pm .6$ | 93.48 | $\underline{0.003 \pm .0}$ | $2.90 \pm .3$ | $1.14e^2$ |
| | DiGress [37][†] | $\mathbf{99.37 \pm .0}$ | $35.31 \pm .0$ | $96.74 \pm .0$ | 77.14 | $\mathbf{1.4e^{-4} \pm .0}$ | $\mathbf{0.07 \pm .0}$ | $6.14e^2$ |
| | Wave-GD (Ours) | $\underline{96.95 \pm .3}$ | $89.79 \pm .5$ | $\underline{98.87 \pm .2}$ | **95.20** | $\underline{0.003 \pm .0}$ | $3.85 \pm .1$ | $1.21e^2$ |

†: results obtained with the official public code.

the molecular graph generation task, we validated its ability to capture the complex dependencies between nodes and edges. Notably, Wave-GD successfully generated plausible graphs while adhering to the chemical valency rule, ensuring the realistic representation of molecular structures.

**Baselines and evaluation metrics.** In this experiment, we compared the results from seven baselines and Wave-GD on a real molecular QM9 dataset. Among the baselines, two are autoregressive flow-based models, namely GraphAF [34] and GraphDF [20], which employ discrete latent variables. Five others are one-shot generative models, which include MoFlow [44], GraphEBM [18], EDP-GNN [25], GDSS [13], and DiGress [37]. Following the standard procedure as in [20, 34], the molecule data were kekulized by the RDKit library [15] and hydrogen atoms were removed.

We evaluated the quality of generated molecular samples with six metrics as in [13]: 1) *Validity* is the fraction of valid molecules that hold chemical rules. In our experiment, we did not apply any valency correction or edge resampling. 2) *Novelty* measures the proportion of valid molecules that are not present in the training set. 3) *Uniqueness* is the proportion of generated molecules that are not duplicated. 4) *Neighborhood subgraph pairwise distance kernel* (NSPDK) [3] is the MMD between the generated molecules and test molecules which considers both the nodes and edges for evaluation. 5) *Fréchet ChemNet Distance* (FCD) [30] evaluates the distance between distributions of the training and generated sets using the activations of the ChemNet layer. 6) *Time* quantifies the duration required to generate 10,000 molecules in the RDKit molecule format. In addition to these metrics, we also report the average values of validity, novelty, and uniqueness to present a unified evaluation over various aspects of the generated graphs.

**Results.** As shown in Table 4, our model outperformed all baselines on the average of validity, novelty, and uniqueness and showed the second-best results on validity, uniqueness, and NSPDK. To be specific, our method showed at most $27.49\%p$ gain over GraphEBM and $18.06\%p$ margin over DiGress in terms of the averaged statistic. Note that, DiGress performed better than our model in NSPDK and FCD, however, its lower novelty (35.31%) inherently leads to lower FCD and NSPDK scores. As the molecular generation task aims to create previously unseen molecular structures, novelty may be a more critical measure on which Wave-GD performed better.

In general, Wave-GD outperformed most of the baselines, excelling not only in generation quality but also in generation speed. It was faster than all autoregressive models, showing $442\times$ speed up compared to GraphDF and $21\times$ speed up compared to GraphAF. Moreover, our method requires a shorter generation time than discrete-time diffusion models. For example, our method was $36\times$ faster than EDP-GNN and $5\times$ faster than DiGress. These results demonstrate that employing a continuous-time diffusion process for converting graphs into noise and vice versa is considerably more efficient compared to the discrete-step noise perturbation utilized in EDP-GNN and DiGress.

### 4.5 Ablation Studies

**Ablation study on oversmoothing.** In Fig. 5, we present the effect of the number of GMH layers (i.e., depth $D$) on the quality of generated samples. With larger $D$, while the sample quality of GDSS is decreased with high averaged MMDs (blue line) due to the oversmoothing, the average MMD of Wave-GD (yellow line) is consistently low representing high fidelity in the generated samples. This is because Wave-GD captures diverse graph frequency characteristics by adaptive multi-resolution filtering, which allows the model to preserve fine-grained details even in deeper layers.

|  | $J$ | Deg. ↓ | Clus. ↓ | Orb. ↓ | Avg. ↓ |
|---|---|---|---|---|---|
| Community-small | 3 | 0.011 | 0.072 | 0.003 | 0.028 |
|  | 4 | 0.017 | 0.070 | 0.007 | 0.031 |
|  | 5 | 0.006 | 0.065 | **0.002** | 0.024 |
|  | 6 | 0.007 | **0.058** | **0.002** | **0.022** |
|  | 7 | **0.005** | 0.069 | 0.009 | 0.027 |
|  | $J$ | Val. ↑ | Uniq. ↑ | Nov. ↑ | Avg. ↑ |
| QM9 | 4 | **97.73** | 79.21 | 92.68 | 89.87 |
|  | 5 | 96.82 | 88.68 | **99.07** | **94.85** |
|  | 6 | 96.70 | **88.95** | 98.91 | **94.85** |
|  | 7 | 96.36 | 88.10 | 98.82 | 94.42 |

Table 5: Ablation studies on the number of scales.

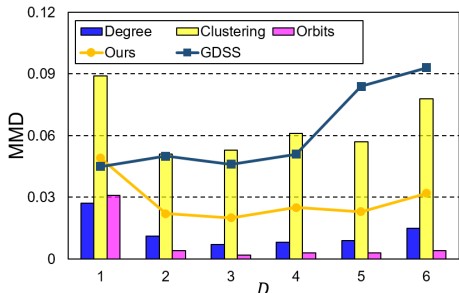

Figure 5: Comparison between Wave-GD (yellow line) and GDSS (blue line) for GMH depths $D$ on Community-small. Bar graphs are MMDs of Wave-GD.

**Ablation study on the number of scales.** In Table 5, we report degree, cluster, orbit MMDs, and their averaged value on the Community-small by changing the numbers of scales $J$. Also, validity, uniqueness, novelty, and their averaged value are presented on the QM9 dataset for multiple $J$'s. All experiments were replicated three times and the averaged results are reported. We observed that the averaged MMD slightly depends on the number of scales, however, the overall quality of generated samples is still better than existing baselines in most settings.

### 4.6 Limitations

Although Wave-GD is robust to varying numbers of scales as shown in Table 5, the choice of $J$ differs across datasets which should be carefully chosen by a user. Also, as Wave-GD performs multi-resolution diffusion, it requires larger resources than conventional models during training. However, the diffusion processes are performed in parallel so there is no overhead in time. Moreover, as sampling is performed without multi-resolution filtering, the generation process is still quite efficient.

## 5 Conclusion

In this paper, we proposed Wave-GD which leverages spectral dependencies between node and edge signals to better characterize their joint distributions via a score-based diffusion model. By capturing their multi-resolution coherence, the model is able to generate graphs of high-fidelity preserving frequency characteristics of the graphs from the training samples. Extensive validation on various datasets and superior performance of Wave-GD highlights its significant potential for various application domains for graph modeling and generation.

**Acknowledgement.** This research was supported by NRF-2022R1A2C2092336 (40%), IITP-2022-0-00290 (40%), IITP-2019-0-01906 (AI Graduate Program at POSTECH, 10%), IITP-2022-2020-0-01461 (ITRC, 10%) from South Korea, and NSF IIS CRII 1948510 from the U.S.

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

# Appendix

In this appendix, we present **1)** the proof of Lemma 1 given in the main paper, **2)** additional implementation details of Wave-GD, **3)** standard deviations of MMDs for generic datasets, **4)** additional ablation studies on the number of scales on Ego-small and Grid datasets, **5)** holistic architecture of two score-based models $\mathcal{X}_\theta$ and $\mathcal{A}_\phi$, and **6)** qualitative comparison of generated samples for all datasets which were not included in the main manuscript due to the page limit.

## A    Proof of Lemma 1

**Lemma 1.** *Let $G$ be an undirected graph, with a node signal $\mathbf{X}$ and a symmetric adjacency matrix $\mathbf{A}$. Given wavelet coefficients $W_\mathbf{A}(s)$ and $W_\mathbf{X}(s)$, which are spectral representations of $\mathbf{X}$ and $\mathbf{A}$ at a specific scale $s$, a dot product of these wavelet coefficients is equivalent to the product between the scale-filtered edge signals and the node signals in the graph space, i.e., $W_\mathbf{A}(s) \cdot W_\mathbf{X}(s) = \mathbf{A}^s \mathbf{X}$.*

*Proof.* With $\psi_s = \mathbf{U}k(s\mathbf{\Lambda})\mathbf{U}^T$ as the basis, the wavelet coefficients of the edge and node signals are given as follows:

$$
\begin{aligned}
W_\mathbf{A}(s) &= \psi_s \cdot \mathbf{A} = \mathbf{U}k(s\mathbf{\Lambda})\mathbf{U}^T\mathbf{A} \\
W_\mathbf{X}(s) &= \psi_s \cdot \mathbf{X} = \mathbf{U}k(s\mathbf{\Lambda})\mathbf{U}^T\mathbf{X}.
\end{aligned}
\tag{10}
$$

As the $k(s\mathbf{\Lambda})$ is a diagonal matrix and $\mathbf{U}$ is a set of orthonormal eigenvectors, the dot product of $W_\mathbf{A}(s)$ and $W_\mathbf{X}(s)$ is computed as

$$
\begin{aligned}
W_\mathbf{A}(s) \cdot W_\mathbf{X}(s) &= (\mathbf{U}k(s\mathbf{\Lambda})\mathbf{U}^T\mathbf{A})^T(\mathbf{U}k(s\mathbf{\Lambda})\mathbf{U}^T\mathbf{X}) \\
&= \mathbf{A}^T\mathbf{U}k^T(s\mathbf{\Lambda})\mathbf{U}^T\mathbf{U}k(s\mathbf{\Lambda})\mathbf{U}^T\mathbf{X} \\
&= \mathbf{A}\mathbf{U}k(s\mathbf{\Lambda})k(s\mathbf{\Lambda})\mathbf{U}^T\mathbf{X} \\
&= \mathbf{A}\mathbf{U}k^2(s\mathbf{\Lambda})\mathbf{U}^T\mathbf{X}.
\end{aligned}
\tag{11}
$$

From Eq. (3) in the main paper, $\mathbf{A}^s = \psi_s \cdot W_\mathbf{A}(s) = \mathbf{U}k^2(s\mathbf{\Lambda})\mathbf{U}^T\mathbf{A}$, and Eq. (11) becomes

$$
\begin{aligned}
W_\mathbf{A}(s) \cdot W_\mathbf{X}(s) &= (\mathbf{A}^s)^T\mathbf{X} \\
&= \mathbf{A}^s\mathbf{X}.
\end{aligned}
\tag{12}
$$

as $\mathbf{A}^s$ is symmetric. Therefore, a dot product of wavelet coefficients of the edge and node signals can be replaced with a spatial graph convolution form in the graph domain by applying an inverse transform only to the edge coefficients. $\qquad\square$

# B  Additional implementation setting

We utilized the PyTorch framework to implement Wave-GD and trained the model using a single NVIDIA GeForce RTX 3090 GPU. In Table 6, we provide details of the implementation settings of Wave-GD. For a fair comparison, we followed the same settings of data splitting and node features as in [13, 17, 25, 42]. For generic datasets, the node features are initialized as a one-hot encoding of the degrees and the generated adjacency matrices are quantized with the operator $1_{x>0.5}$ to obtain binary edges. As in [13], we used two types of SDEs: Variance Preserving (VP) SDE and Variance Exploding (VE) SDE [36] for diffusion processes. To solve the SDEs, we used the predictor-corrector sampler (PC sampler) [36] with either the Euler-Maruyama (EM) predictor or the reverse diffusion predictor (Rev), which discretizes the reverse-time SDE. As the corrector for the PC sampler, Langevin MCMC [6, 27] was employed.

Table 6: Hyperparameters of Wave-GD for all datasets. We provide hyperparameters and detailed settings for training two score-based models $\mathcal{X}_\theta$ and $\mathcal{A}_\phi$ and diffusion processes with SDE.

|  | Hyperparameter | Ego-small | Community-small | Grid | QM9 |
|---|---|---|---|---|---|
| $\mathcal{X}_\theta$ | Number of attention heads | 4 | 4 | 4 | 4 |
|  | Number of initial channels | 2 | 2 | 2 | 2 |
|  | Number of hidden channels | 8 | 8 | 8 | 8 |
|  | Number of final channels | 4 | 4 | 4 | 4 |
|  | Number of GCN layers | 5 | 5 | 7 | 3 |
|  | Hidden dimension | 32 | 32 | 32 | 16 |
| $\mathcal{A}_\phi$ | Number of scales $J$ | 11 | 6 | 4 | 6 |
|  | Number of attention heads | 4 | 4 | 4 | 4 |
|  | Number of initial channels | 2 | 2 | 2 | 2 |
|  | Number of hidden channels | 8 | 8 | 8 | 8 |
|  | Number of final channels | 4 | 4 | 4 | 4 |
|  | Number of GCN layers | 5 | 5 | 7 | 3 |
|  | Hidden dimension | 32 | 32 | 32 | 16 |
| SDE for $\mathbf{X}$ | Type | VP | VP | VP | VE |
|  | Number of sampling steps | 1000 | 1000 | 1000 | 1000 |
|  | $\beta_{min}$ | 0.1 | 0.1 | 0.1 | 0.1 |
|  | $\beta_{max}$ | 1.0 | 1.0 | 1.0 | 1.0 |
| SDE for $\{\mathbf{A}^{s_i}\}_{i=0}^J$ | Type | VP | VP | VP | VE |
|  | Number of sampling steps | 1000 | 1000 | 1000 | 1000 |
|  | $\beta_{min}$ | 0.1 | 0.1 | 0.2 | 0.1 |
|  | $\beta_{max}$ | 1.0 | 1.0 | 0.8 | 1.0 |
| Solver | Type | EM | EM + Langevin | Rev. + Langevin | Rev. + Langevin |
|  | SNR | - | 0.05 | 0.1 | 0.2 |
|  | Scale coefficient | - | 0.7 | 0.7 | 0.7 |
| Train | Optimizer | Adam | Adam | Adam | Adam |
|  | Learning rate | $8 \times 10^{-4}$ | $5 \times 10^{-3}$ | $1 \times 10^{-3}$ | $2 \times 10^{-3}$ |
|  | Weight decay | $1 \times 10^{-4}$ | $1 \times 10^{-4}$ | $1 \times 10^{-4}$ | $1 \times 10^{-4}$ |
|  | Batch size | 128 | 128 | 8 | 1024 |
|  | Number of epochs | 5000 | 5000 | 6000 | 300 |
|  | Exponential Moving Average | - | - | 0.999 | - |

## C   Generation results with standard deviation on generic datasets

In Table 1 and Table 3 of the main manuscript, we reported the mean MMDs of three independent runs on generic datasets, using different model parameter initializations. In Table 7, we provide the standard deviation of degree, clustering, and orbit MMDs, as well as the time required to generate samples for each generic dataset. The generation time slightly increases in the experiments where more samples (i.e., 1024 samples) are generated on Ego-small and Community-small datasets. Furthermore, as the graph becomes more complicated with more nodes and edges, the generation time also increases accordingly.

Table 7: Generation results of Wave-GD on the generic datasets. The results are the mean MMDs of three runs and their standard deviation.

| # of test data | # of samples | Degree | Cluster | Orbit | time (s) |
|---|---|---|---|---|---|
| **Ego-small** | | | | | |
| 40 | 40 | $0.012 \pm 0.001$ | $0.010 \pm 0.001$ | $0.005 \pm 0.001$ | 40.03 |
| | 1024 | $0.010 \pm 0.001$ | $0.018 \pm 0.003$ | $0.005 \pm 0.002$ | 40.95 |
| **Community-small** | | | | | |
| 20 | 20 | $0.007 \pm 0.002$ | $0.058 \pm 0.004$ | $0.002 \pm 0.0$ | 89.80 |
| | 1024 | $0.016 \pm 0.003$ | $0.077 \pm 0.006$ | $0.001 \pm 0.002$ | 94.75 |
| **Grid** | | | | | |
| 20 | 20 | $0.144 \pm 0.004$ | $0.004 \pm 0.001$ | $0.021 \pm 0.001$ | 733.79 |

## D   Ablation studies on the number of scales

Along with Table 5 in the main paper, we report the ablation studies on the number of scales $J$ for Ego-small and Grid datasets in Table 8. All experiments with different $J$ were replicated three times and their averaged results were reported. We found that the average MMDs on the Ego-small slightly differ depending on the $J$ and that of Grid is relatively consistent along the $J$'s. In general, the number of scales affects the sample quality depending on the datasets, however, the averaged MMD of Wave-GD is still better than existing methods in most settings.

Table 8: Ablation studies on the number of scales.

| | Ego-small | | | | | Grid | | | |
|---|---|---|---|---|---|---|---|---|---|
| $J$ | Deg. | Clus. | Orb. | Avg. | $J$ | Deg. | Clus. | Orb. | Avg. |
| 8 | 0.019 | 0.016 | 0.011 | 0.015 | 3 | **0.142** | 0.003 | 0.034 | 0.059 |
| 9 | 0.016 | **0.001** | 0.015 | 0.011 | 4 | 0.144 | 0.004 | 0.021 | **0.056** |
| 10 | 0.015 | 0.008 | **0.005** | **0.009** | 5 | 0.161 | **0.001** | **0.015** | 0.059 |
| 11 | **0.012** | 0.010 | **0.005** | **0.009** | 6 | 0.168 | 0.005 | **0.015** | 0.062 |
| 12 | 0.018 | 0.016 | 0.013 | 0.016 | - | - | - | - | - |

# E    The architecture of the score-based models of Wave-GD

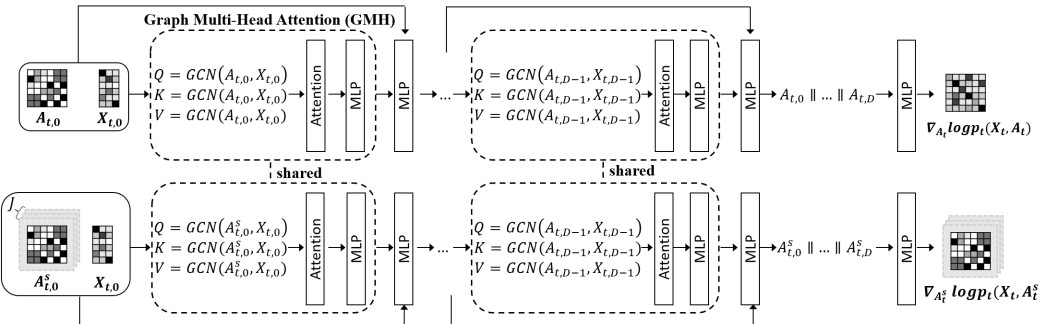

Figure 6: The architecture of $\mathcal{X}_\theta$ that estimates the partial score $\nabla_{\mathbf{X}_t} \log p_t(G_t)$.

Figure 7: The architecture of $\mathcal{A}_\phi$ that estimates the partial scores $\nabla_{\mathbf{A}_t} \log p_t(G_t)$ and $\{\nabla_{\mathbf{A}_t^{s_i}} \log p_t(G_t^{s_i})\}_{i=1}^{J}$.

Here we illustrate the architecture of score-based models $\mathcal{X}_{\theta,t}$ and $\mathcal{A}_{\phi,t}$ in Figure 6 and Figure 7, respectively, which are described in Section 3.5 in the main manuscript. Basically, we utilized graph multi-head attention (GMH) [1] as in GDSS [13], which is an iterative attention mechanism using graph convolutions as query, key, and values.

Given $H_{t,0} = \mathbf{X}_t$, the attention operation GMH$(\cdot)$ is performed $D$ times to estimate the partial scores $\nabla_{\mathbf{X}_t} \log p_t(\mathbf{X}_t, \mathbf{A}_t)$ and $\{\nabla_{\mathbf{A}_t^{s_i}} \log p_t(\mathbf{X}_t, \mathbf{A}_t^{s_i})\}_{i=0}^{J}$ as follows:

$$\mathcal{X}_{\theta,t}(G_t) = \text{MLP}_\theta([\{\text{GMH}_\theta(H_{t,d}, \mathbf{A}_{t,d})\}_{d=0}^{D}]),$$
$$\mathcal{A}_{\phi,t}(G_t^{s_i}) = \text{MLP}_\phi([\{\text{GMH}_\phi(H_{t,d}, \mathbf{A}_{t,d}^{s_i})\}_{d=0}^{D}]), \tag{13}$$

respectively, where [·] represents the concatenation of GMH outputs and $H_{t,d+1} = \text{GNN}(H_{t,d}, \mathbf{A}_{t,d})$.

As shown in Figure 7, the score-based model $\mathcal{A}_\phi$ is shared across the multi-resolution edges and the given adjacency matrices. During *training*, it estimates the partial scores in parallel along the $J+1$ graphs using a graph convolution with node features $\mathbf{X}$ in the GMH layers, so that the model is able to capture generalized spectral coherence across different resolutions. However, in the *sampling* process, only the partial score of $\mathbf{A}_t$ (i.e., $\nabla_{\mathbf{A}_t} \log p_t(\mathbf{X}_t, \mathbf{A}_t)$) is estimated from $\mathcal{A}_\phi$ as generating samples of multi-resolution edges is not our aim. For the other score-based model $\mathcal{X}_\theta$, it produces the partial score $\nabla_{\mathbf{X}_t} \log p_t(\mathbf{X}_t, \mathbf{A}_t)$ in both training and the sampling processes as it does not rely on multi-resolution approach.

# F Qualitative evaluation of generated samples

## F.1 Comparison with state-of-the-art baselines on generic datasets

In Figures 8, 9, and 10, we qualitatively compare the generation results of Wave-GD on the generic datasets with two state-of-the-art diffusion models for graph generation: DiGress [37] and GDSS [13]. The visualized graphs demonstrate the superiority of Wave-GD which closely simulates training data compared to the SOTA baselines.

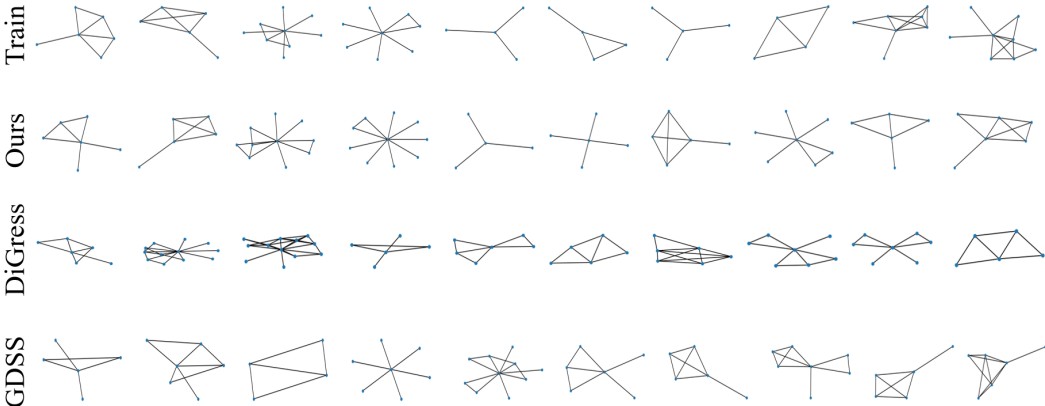

Figure 8: Non-curated samples generated by Wave-GD, DiGress, and GDSS trained on the Ego-small dataset.

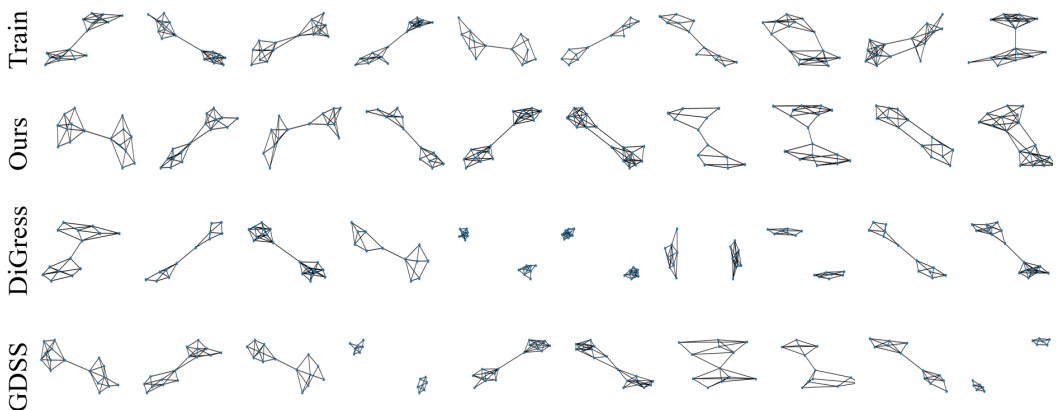

Figure 9: Non-curated samples generated by Wave-GD, DiGress, and GDSS trained on the Community-small dataset.

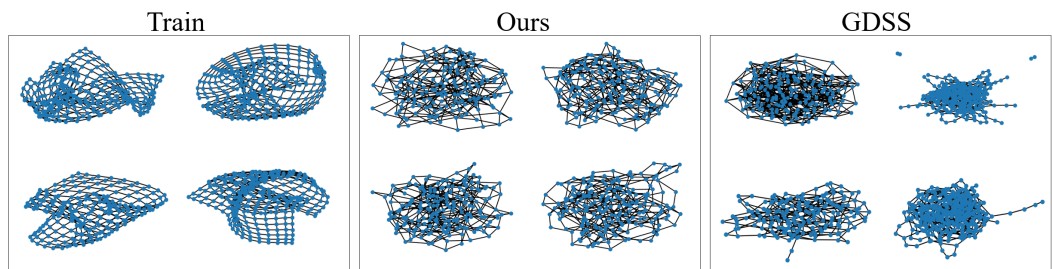

Figure 10: Non-curated samples generated by Wave-GD, DiGress, and GDSS trained on the Grid dataset.

## F.2 Comparison with state-of-the-art baselines on QM9 dataset

In Figure 11, we provide the visualization of training datasets and generated samples of Wave-GD for the molecular dataset. Along with the visualization, a quantitative comparison is also provided by measuring Tanimoto similarity based on Morgan fingerprints, which are calculated by the RDKit library [15]. For each generated molecule, the pairwise similarity between the training sample is calculated. As shown in the figure, Wave-GD successfully generates high-quality molecules that closely resemble the training data, whereas other baseline methods produce molecules that deviate from the training distribution.

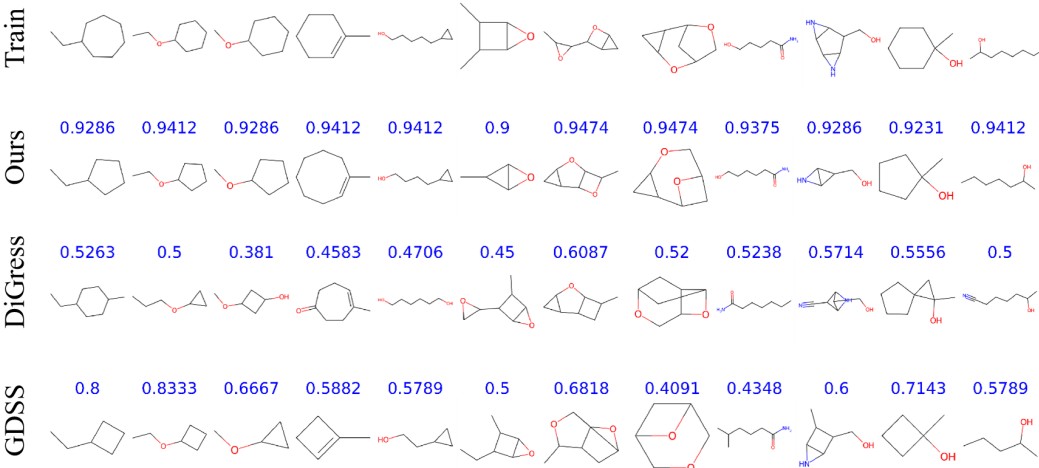

Figure 11: Visualization of the sampled graphs on the QM9 dataset with maximum Tanimoto similarity. For each graph, we display the one-to-one similarity to the real training samples at the same location in the upper panel.

