# OpenReview forum: "Multi-resolution Spectral Coherence for Graph Generation with Score-based Diffusion"
_NeurIPS.cc/2023/Conference — NeurIPS 2023 poster_

### Official Review · Reviewer_ZNjR · 2023-06-25

**Soundness:** 2 fair
**Presentation:** 3 good
**Contribution:** 2 fair
**Rating:** 6
**Confidence:** 3

**Summary:**

This paper proposes Wave-GD, a score-based generative model, to generate graphs with high fidelity. By capturing the dependency between nodes and edges at multiple resolutions in the spectral space, it claims it overcomes the over-smoothing problem and achieves real-like frequency characteristics of nodes and edges.

**Strengths:**

1. The paper is well-written and easy to follow.
2. The paper's motivation is clear, and the paper has shown some limitations that previous models may have.
3. The idea of the paper is simple and straightforward.

**Weaknesses:**

1. The experiment is not sufficient to support the proposed method.

(1) lack of baselines: some baselines that utilize spectral information/graph characteristics are missing. It would be great to see the author compare them.[1,2,3]

(2) lack of more robust graph datasets: Ego-small and Community-small graphs are too simple, I think over-smoothing can't be a major issue for graphs in such scales. And how the "multi-resolution" should be defined in such small graphs? I suggest the authors run experiments on more complex graphs such as Planar graphs, SBM graphs, and large networks if possible with hundreds of nodes.

(3) I suggest the author also include DiGress in Figure 5 -- since it also utilizes spectral information.

(4) Another comment in Figure 5: it's not clear whether the performance drop when increasing #GNN layers is really due to the over-smoothing problem, more investigation should be conduct to further prove this phenomenom.

2. While the author propose to also diffuse on the edge weights obtained from SGWT, there are also other ways to define the edge importance (e.g., edge conductance). Justiifcation should be provided why SGWT is the chosen over others.

[1] Luo, Tianze, Zhanfeng Mo, and Sinno Jialin Pan. "Fast Graph Generative Model via Spectral Diffusion." arXiv preprint arXiv:2211.08892 (2022).

[2] Martinkus, Karolis, et al. "Spectre: Spectral conditioning helps to overcome the expressivity limits of one-shot graph generators." International Conference on Machine Learning. PMLR, 2022.

[3] Chen, Xiaohui, et al. "Efficient and Degree-Guided Graph Generation via Discrete Diffusion Modeling." arXiv preprint arXiv:2305.04111 (2023).

**Questions:**

Please address the weakness above and:

1. How exactly can the SGWT addresses the over-smoothing problem?

2. Can you give a concrete example why it can capture "multi-resolution" of the graph, and can you provide a formal definition of the multi-resolution.

3. what's the motivation of conducting the experiment with larger sample size?

**Limitations:**

1. The result on molecule generation is not that expressive.

2. The contribution may be insignificant: there are many GNN design that may alleviate the over-smoothing problem. Any baseline model with a better score networks may overcome this limitation easily.

---

> ### Author Rebuttal · Authors · 2023-08-08
>
> W1) Compare with more baselines.
>
> A) We thank the reviewer for introducing great references. We compared Wave-GD with Martinkus et al (ICML 2022) and Chen et al (ICML 2023) and reported the results in the pdf of the global response. As shown in the result, our method outperformed both baselines in three datasets. We will add these results in the revision and discuss their methods.
>
> W2) Compare with additional datasets with larger graphs.
>
> A) As suggested by the reviewer, we conducted an experiment on the Planar data and presented both qualitative and quantitative results in the global response. We compared MMDs of test and generated graph sets on degree, clustering coefficients, and orbit counts. Our method outperformed five baselines including SPECTRE and EDGE for the average of three MMDs.
> The resultant optimal scales for the Planar dataset were also reported in the general response, which shows that their optimal scales are relatively smaller than other generic graph datasets. These results indicate that capturing high-frequency and localized features from individual nodes was more informative than capturing cluster-related features in this dataset.
>
> Note that we rushed these experiments with limited time and resources, and hyperparameters such as learning rate and $J$ were not fine-tuned. But still, we obtained the results of Wave-GD outperforming many recent baselines. Regarding SBM dataset, we were not able to pull reportable results given limited time but confirmed them to be promising. We plan to prepare a journal version and these additional experiments in more complete form will be included there.
>
> W3) Include DiGress in Figure 5.
>
> A) We thank the reviewer for the suggestion. However, in Figure 5, we compared our method with GDSS for multiple Graph Multi-Head attention (GMH) layers to observe robustness against oversmoothing caused by repetitive graph convolutions (within the GMH layers). As DiGress does not use graph convolutions nor GMH layers, we cannot directly compare our method with DiGress to examine its performance against oversmoothing problem, we believe that the reviewer will easily see why DiGress cannot be compared directly within Figure 5.
>
> W4/Q1) It is not clear that the performance drop in Figure 5 is caused by the increase in the number of GNN layers. How can the SGWT address the over-smoothing problem?
>
> A) The performance drop in Figure 5 is caused by repetitive graph convolutions with increasing GMH layers. As shown in lines 190-192, the query, key, and values in the GMH layer were made up of graph convolutions, which aggregate features from neighboring nodes. Many previous literatures [1, 2] have shown that repetitive graph convolutions with multiple layers cause the oversmoothing problem.
>
> Both our method and GDSS used GMH layers and graph convolutions, however, SWGT in our method allows a model to flexibly preserve discriminative characteristics of graph representations. This is because filtered eigenvalues $k(s\Lambda)$ and corresponding eigenvectors restrict the extent of message propagation and preserve the unique characteristics of localized signals. This spectral filtering with limited eigenvectors/eigenvalues prevents the over-smoothing issue in deep layers compared to using unfiltered raw data.
>
> We conducted an empirical analysis to support this claim by comparing Mean Average Distance (MAD) [1] between $AX$ and $A^sX$, where the MAD is a metric to measure the smoothness of the graph representation. Given a template graph shown in Fig. 1a in the main paper, $X_{i+1} = AX_{i}$ (i=0,1,2) was calculated as in 3-layer graph convolution (without weights), where the $X_0$ is a one-hot encoded degree matrix. The MAD value of $AX_3$ was 0.079 and that of $A^sX_3$ with s=20 was 0.724, which is 9 times larger. The low MAD value without SGWT represents that the node representations become indistinguishable, and the higher MAD with SGWT shows that the local and discriminative features were preserved.
>
> Q2) Provide an example and definition of multi-resolution.
>
> A) The multi-resolution is a hierarchical concept [3] to capture information of data at varying levels of granularity. This multi-resolution is well-established with Wavelet transform (Mallat, 1999) and Spectral Graph Wavelet Transform (Hammonds, 2012) extends its concept to graphs as which is utilized in our framework. Basically, it is the band-pass filter $k(s)$ which covers different bands, i.e., scales, in the frequency space. Given a signal $x$, controlling $s$ in Eq (1) and (2) in the main paper yields the representation of $x$ in different resolutions. We will clarify these in the preliminary section of our revision.
> For empirical evaluation, in Sec. 7 of supplementary material and the general response, we provided the value of the actual converged scales for generic graph datasets for our experiments, which show the capability of Wave-GD to capture different multi-resolution graph representations considering the characteristics of datasets.
>
> Q3) What is the motivation for the experiment with larger samples?
>
> A) As in EDP-GNN [4] and GDSS [5], we performed the experiment with more samples to extensively assess the quality of generated samples. With more data, the larger sample set may contain more abnormalities or high-fidelity samples. Therefore, the experiment aims to assess how the quality of generated samples changes with greater diversity.
>
> [1] Chen et al. "Measuring and relieving the over-smoothing problem for graph neural networks from the topological view." AAAI. 2020.
>
> [2] Zhao and Akoglu. "Pairnorm: Tackling oversmoothing in GNNs." ICLR, 2020.
>
> [3] Rosenfeld, “Multiresolution image processing and analysis.” Springer Science & Business Media, 2013.
>
> [4] Niu et al. “Permutation invariant graph generation via score-based generative modeling.” AISTATS, 2020
>
> [5] Jo et al. “Score-based generative modeling of graphs via the system of stochastic differential equations.” ICML, 2022

---

> > ### Comment · Reviewer_ZNjR · 2023-08-14
> >
> > My concerns are mostly addressed, I'd like to raise my score

---

### Official Review · Reviewer_nurq · 2023-07-05

**Soundness:** 3 good
**Presentation:** 2 fair
**Contribution:** 3 good
**Rating:** 6
**Confidence:** 4

**Summary:**

In this work the authors tackle the problem of graph generation learning where the goal is to learn the key features of a set of graphs and be able to generate graphs with similar properties. To that extend, the authors extend the GDSS (Jo 2022) method through an additional loss term (Eq. 7). This loss term encourages the employed GNNs to learn to reconstruct spectrally modified matrices A^s_i in addition to the normal adjacency matrices. The spectrally modified matrices are obtained through an SVD/PCA like approach where some spectral properties of the adjacency matrix are accentuated. This spectral accentuation is learnable as part of the training procedure.
The authors evaluate their approach on two real world and two synthetic datasets also used in previous studies. In terms of MMD (Mean maximum deviation) their approach frequently outperforms other methods employed for the task of graph generation learning. On molecule data, the employed procedure still ranks amount the best.


**Strengths:**

The authors provide an interesting extension to an existing approach (GDSS), which allows to better learn the scales that are important for a specific set of graphs. The greatly increased stability during training compared to GDSS (Figure 3) seems promising. The paper contains extensive comparison to other methods. The presented approach is on par/outperforms these other methods on the presented datasets. Most computational overhead is in the training phase, inference is as fast as GDSS.

**Weaknesses:**

Overall the core weakness of the paper is its presentation, which has a lot of room for improvements and weak support for the main claims of the paper (last point).

The presentation of the SDE learning (lines 155-179) is not understandable without reading the GDSS paper. I think it could be drastically improved by highlighting the differences/ improvements in comparison to GDSS rather than repeating all the definitions/equations.

The description of Figure 2a) could be much improved as it is not well understandable without reading the rest of the paper first.

Some of the equations and notation does not further the main cause of the paper: Section 3.1 beyond the first paragraph can almost entirely be cut. The introduced transformation $X^{s_i}$ on the edge signal X is not really used.

Lemma 1 seems to be misplaced as it introduced (lines 138ff) quite far away from where it is used (lines 190ff). Also it seems Lemma 1 is likely not a new result.

The bold highlights in Table 1 are not correct, in column “orbit”, three other methods outperform the presented approach but are not highlighted in bold. Also in lines 289f the text mismatches the table.

It is also unclear from the text how much hyperparameter tuning was done to achieve the results presented.

Last but not least, the experimental section shows, from an applicant's point of view, that the proposed changes lead to empirical performance increase on the graph generation task. On the other hand the experimental section does not well support whether the introduced changes actually had the desired effect of learning different “scales” better.

It might as well be, that the introduced changes simply increase training robustness. More on this in the “Questions” section. It seems that one would need additional support for the main claim.

**Questions:**

It would be interesting to see which scales lambda (as in lines 219, 220) have actually been obtained from the training. Do they differ significantly for different datasets? Would an equidistant choice of lambda be sufficient?
Similarly it would be interesting to see the relative strengths of the lambda_A/ lambda_A^{s_i} (as in eq.7). From a theoretical perspective (thinking of spectral clustering) one would assume that mostly low frequency modes are relevant for the community dataset, is that really the case?
Lastly, it would be interesting to see whether the performance increase obtained from the spectrally filtered adjacency matrix is any better than just introducing A^{s_i}

**Limitations:**

see questions above

---

> ### Author Rebuttal · Authors · 2023-08-08
>
> W1) Improve clarity on SDE learning, Fig. 2a, Sec 3.1, Lemma 1, and Tab. 1.
>
> A) We thank the reviewer for the detailed constructive comments.
> We presented a revised version of Fig. 2a with its improved description in the pdf of the general response, so please check on it. We will also clarify all other places based on the reviewer’s suggestions, such as highlighting the difference between our method and GDSS by adding the description of the loss in Eq. 7 and modifying the bolds in Table 1 in the revision.
>
> Specifically, we will add the following descriptions in line 180, before the Remark section: “Both our method and GDSS perform denoising score-matching to the partial scores of $X$ and $A$. However, our method additionally models the joint probability space of $X$ and $A^s$ via SDEs. This is realized by the loss in Eq. (7), and this operation allows a model to flexibly estimate the complex dependency between nodes and edges with multi-resolution SGWT. Also, note that the scales of SWGT {$s_i$}$_{i=1}^J$ in Eq. (7) are trainable so that multi-level granularities that characterize graph distribution can be adaptively captured during training.”.
>
> For the $X^s$ in Sec 3.2, it is not used in our method, however, it can be used by replacing $A^sX$ with $AX^s$. As shown in the proof of Lemma 1 in the supplementary material, $W_{A}(s) \cdot W_{X}(s) = AUk^2(s\Lambda)U^TX = A^sX = AX^s$. Therefore, either $A^sX$ or $AX^s$ can be used to capture the spectral coherence between node features and graph structures. We will add this explanation to the revised manuscript for better understanding.
>
> W2) Lemma 1 is likely not a new result.
>
> A) To our best knowledge, Lemma 1 and its proof are novel ideas as it proposes that computing the coherence as a dot product of multi-resolution nodes and edges in the spectral space is equivalent to a graph convolution in the original graph space. Please let us know of any references that suggested the same idea so that we can properly credit them.
>
> W3) How much hyperparameter tuning was done?
>
> A) Basically, we followed most hyperparameter settings of GDSS as Wave-GD is built upon the GDSS. The hyperparameter we mainly tried out was learning rate and $J$, i.e., the number of scales. As we used two wavelet filters (i.e., low-pass and band-pass filters), we conducted experiments with at least two scales ($J=2$). Subsequently, we increased $J$ by 1 for the band-pass filter until the optimal results were achieved. For the Grid dataset, the maximum $J$ we could try was $6$ due to computational limitations. We provided ablation studies on $J$ in Table 4 in the main paper and Section 4 in the supplementary material for all datasets. While the optimal $J$ may be different across different datasets, the results within each dataset are not very sensitive but rather robust to the choice of $J$. For the learning rate, we performed a grid search in {$0.0008, 0.001, 0.002, 0.005, 0.01$} for each dataset.
>
> W4/Q1) Which scales were obtained? The values should be presented in the experimental section.
>
> A) In the pdf of the global response, we reported the table of converged scale values. We will add the table in the revised paper.
>
> In Section 7 of the supplementary material, we provided all scale convergence flows along epochs and analyzed the converged scale ranges for generic graph datasets. We observed that the scales differ for different datasets. Interestingly, the scales of the large Grid dataset (with $100  \leq |V| \leq 400$ nodes) were generally smaller compared to the Ego-small ($4 \leq |V| \leq18$) and Community-small ($12 \leq |V| \leq 20$) datasets. Specifically, the converged scales of Ego-small ranged within $[20, 47]$ and the scales of Grid ranged within $[12, 36]$. Note that the smaller scales capture local graph features with higher frequencies. Therefore, the results demonstrate that capturing local and detailed graph representations is more critical when dealing with complex and large graphs, in comparison to smaller graphs. In other words, for the Community-small dataset, features in relatively low frequency (e.g., cluster-related features) were captured with larger scales.
>
> Q2) Relative strength of lambda in Eq. 7.
>
> A) $\lambda$'s in Eq. 7 were set to 1 so the relative strength between $A$ and $A^{s}$ were the same.
>
> Q3) Performance comparison between spectrally filtered adjacency matrix and $A^{s_i}$.
>
> A) The use of spectrally filtered adjacency matrix (i.e., wavelet coefficient $W_{A}(s)=\psi_s \cdot \mathbf{A}$) with spectrally filtered node features (i.e., $W_{X}(s)$) will make the same result as using $A^s$ and $X$. As shown in the proof of Lemma 1 in the supplementary material, $W_{A}(s) \cdot W_{X}(s) = A^{s}X$, and therefore the result of using either side of the data will be equivalent.

---

> > ### Comment · Reviewer_nurq · 2023-08-14
> >
> > I thank the authors for their detailed response. In light of the responses by the authors I have increased my score.

---

### Official Review · Reviewer_jRF4 · 2023-07-06

**Soundness:** 3 good
**Presentation:** 2 fair
**Contribution:** 3 good
**Rating:** 6
**Confidence:** 3

**Summary:**

The paper introduces a graph generative approach that leverages diffusion models and wavelet theory. The key concept revolves around utilizing the wavelet transform of the adjacency matrix across various scales, and learning a joint backward diffusion process that remains valid at all considered scales simultaneously. Consequently, the proposed approach exhibits a multi-resolution characteristic. The proposed approach is evaluated in the graph generation task using four benchmarks and is compared against autoregressive and one-shot approaches from previous works.

**Strengths:**

The adaptation of diffusion models to the graph generation field is a timely and intriguing topic, considering the challenges posed by the discrete nature of graph data.

The experimental evaluation provides substantial support for the claims made in the paper. The proposed approach outperforms recent methods on three benchmark in the graph generation task and achieves comparable results in molecule generation task.
The experimental setup and the metric considered for the evaluation are clearly presented.

**Weaknesses:**

Having to choose the parameter *J* without any insight or guidance for different datasets can be a drawback in practical applications. I would suggest to further investigate the relationship between performance and *J* among different datasets, along with the impact of graph statistics on the optimal *J*. Also, it is not clear which how many scales have been used to obtain the results reported in Tables 1-3.

The time complexity of the proposed approach, as indicated in Table 3, could present challenges in certain settings. Further discussion or potential mitigation strategies for addressing this issue would be beneficial.

Clarity could be improved in certain sections. For instance, Figure 1's purpose is unclear to me, and the role of spectral coherence between nodes and edges in the proposed approach needs better explanation. Providing a clear definition of the scale (s) domain and kernel function (k) in the preliminary sections would also help readers understand the concepts more quickly.

**Questions:**

It would be helpful if the authors could provide clarification on why only $A^s$ and not $X^s$ is considered for learning the diffusion process.

Regarding Table 4, should the lower numbers indicate better results? In the QM9 part of the table, higher numbers are bold. Is my interpretation wrong?

The following paper, which explores diffusion in the wavelet coefficient space for 3D shape generation, could be an interesting reference: \
_Hui, Ka-Hei, et al. "Neural wavelet-domain diffusion for 3d shape generation." SIGGRAPH Asia 2022 Conference Papers. 2022_

**Limitations:**

The authors adequately addressed the limitations.

---

> ### Author Rebuttal · Authors · 2023-08-08
>
>
> W1) Ablation study on $J$ and analyses of the impact of graph statistics on the optimal J are needed. Also, how the J’s were set to obtain the main results?
>
> A) We provide our answers in the following, and we will discuss them in the main manuscript to make the paper more clear.
> * We agree that the number of scales $J$ needs to be carefully chosen for each dataset. We already have done the ablation studies on $J$ which are given in Table 4 of the manuscript for Community-small and QM9 datasets and Section 4 of the supplementary for Ego-small and Grid datasets. We observed that the results are not too sensitive to $J$, but we had to try varying numbers to obtain the optimal result.
>
> * For the relationship between $J$ and graph statistics, we empirically observed that relatively smaller $J$ ($J=4$) performs better on large Grid graphs (with $100 \leq |V| \leq 400$) and larger $J$ ($6 \leq J \leq11$) showed the best performances on smaller graphs such as Ego-small ($4 \leq |V| \leq18$), Community-small ($12 \leq |V| \leq 20$) and QM9 ($1 \leq |V| \leq 9$).
>
> * As we mentioned in line 224, to obtain the results in Table 1-3, $J$ was set to $11$, $6$, $4$, and $6$ for the Ego-small, Community-small, Grid, and QM9 datasets, respectively. Also, we analyzed the convergences of $J$ scales along epochs for each dataset and reported the results in Section 7 of the supplementary material. The exact trained scale values were reported in the pdf of the general response.
>
> W2) Time complexity could present challenges.
>
> A) As in GDSS, PC sampling was used with a reverse diffusion predictor and Langevin MCMC corrector for the QM9 dataset. By omitting the correction step and using a predictor-only method, the sampling time was reduced from 154s to 61s. However, we observed a trade-off between sampling time and sample quality, as validity slightly decreased (~1.7%p drop) for a predictor-only method. Regarding the complexity challenge for the spectral decomposition of large graphs, as we mentioned to Reviewer 6cNy, there are approximations available.
>
>
> W3) Need to improve clarity (e.g., Figure. 1, description of the spectral coherence, scales, and kernel function).
>
> A) Thank you for the suggestion. We will improve the clarity that the reviewer pointed out such as a description of the kernels we used, and the definition of scales in the preliminary section as well as in Figure 1. The intention of Figure 1 was to show that certain connections/disconnections are accentuated at specific scales which can be better characterized in the diffusion model, and we will connect dots between Figure 1 and the design of our model to help future readers.
>
>
> Q1) Why only $A^s$ is used and $X^s$ is not considered?
>
> A) As shown in the proof of Lemma 1 in the supplementary material, using either $A^s$ or $X^s$ is enough to capture the spectral coherence between node features and graph structures. Specifically, the Eq. (2) and (3) in the proof show that $W_{A}(s) \cdot W_{X}(s) = AUk^2(s\Lambda)U^TX = A^sX = AX^s$. Therefore, either $A^s$ or $X^s$ should be used with an unfiltered counterpart. We will add this explanation to the revised manuscript for better understanding.
>
>
> Q2) Interpretation in Table 4 is confusing.
>
> A) As in Table 3, higher numbers are better results for QM9. Note that, the metrics we used for the molecular QM9 dataset, i.e., validity, uniqueness, novelty, are different from those for generic graph datasets. Unlike QM9, lower MMD values indicate better results for the generic graph datasets such as Ego-small, community-small, and Grid. We will improve the clarity in Table 4 by adding up/down arrows beside the metrics.
>
> Q3) Suggestion on additional reference.
>
> A) Thank you for suggesting an interesting reference on 3D generative modeling with wavelet representation. The proposed reference and our work basically use different wavelet construction, and we will discuss it in the Related work section.

---

> > ### Comment · Reviewer_jRF4 · 2023-08-21
> >
> > I thank the authors for their time and effort in answering my questions.
> > After having considered both the author's response and the other reviews, I maintain my original score.

---

### Official Review · Reviewer_6cNy · 2023-07-07

**Soundness:** 3 good
**Presentation:** 3 good
**Contribution:** 3 good
**Rating:** 6
**Confidence:** 3

**Summary:**

This paper claims the node feature and graph topology are not coherent in most previous generative graph models and high-frequency signals in node features and graph topology may neglect during the generation process. Therefore, they propose a Wavelet graph diffusion model (Wave-GD) with score-based diffusion. Specifically, it uses different graph wavelet bases to get graph signals in different frequency ranges. The overall model diffuses node features, the original graph, and the adjacency matrix constructed by graph wavelet bases. To improve the coherence between node features and graph topology, the score-based models are based on graph multi-head attention layers, which take the product of node features and adjacency matrices including the original one and ones learned by different bases, which alleviates the gap between node features and graph structures. Performance on three small synthetic datasets and one real-world molecular dataset show that the proposed model can generate graphs that are not only realistic graphs in shape but also obey the chemical rules in high fidelity.

**Strengths:**

1.	The novel part is the paper considers high frequency and discovers the coherence between node features and graph structures in diffusion models but still uses a simple dot product to solve it.
2.	The empirical results on real-world molecular datasets show the effectiveness in terms of generating realistic graphs with high fidelity and high novelty in a relatively fast time.
3.	The model enjoys high flexibility regarding different tasks where they need frequency graph signals at different scales.


**Weaknesses:**

1.	"nodes and edges" in line 26 is misleading, it would be better if you mention "node" means "node features" and "edges" means "graph structures" in the introduction and then use "node" and "edge" for simplicity.
2.	Multi-resolution and coherence are two major claims in your paper, but it is not verified that coherence improves performance.
3.	the proposed model is limited to small graphs due to the decomposition in spectral graph wavelet bases.

**Questions:**

Q1: Would you give some experimental analysis that can show if the main contribution comes from coherence or multi-resolution?


Q2: how do you define $s$ for graph wavelet bases?


**Limitations:**

Yes. It may produce molecules that may harm the body.

---

> ### Author Rebuttal · Authors · 2023-08-08
>
> W1) Change descriptions of the node and edge in line 26.
>
> A) We will use the terms ‘node features’ and ‘graph structure’ in the intro properly as the reviewer suggested. We appreciate the reviewer's comment.
>
> W2/Q1) Does performance improvement come from coherence or multi-resolution?
>
> A) As we have shown in the proof of Lemma 1 (in the supplementary material), the coherence as a dot product between wavelet representation $W_{A}(s)$ and $W_{X}(s)$ at scale $s$ in the spectral domain is equivalent to a simple matrix multiplication $A^{s}X$ in the graph domain. Although the concept of coherence was not used in the previous literature, GDSS used the formulation $AX$ to compute scores for joint distribution learning, and we are essentially deriving its multi-resolution representations and using them as the multi-resolution coherence. Therefore, we conclude that it is the multi-resolution that improves graph generation.
>
> W3) The method is limited to small graphs due to spectral decomposition.
>
> A) We agree that online spectral decomposition can be challenging for extremely large graphs due to its computational costs, i.e., $O(N^3)$ with $N$ nodes. To mitigate this issue, first of all, conventional polynomial approximations for the transforms can be easily achieved based on [1,2], which have been cited in the original manuscript. Moreover, for exact computation, the decomposition does not necessarily have to be performed online. It can be done in the data preprocessing stage to save all eigenvectors and eigenvalues before model training, which can be a reasonable strategy for a population of graphs with relatively small sizes.
>
> To empirically assess the computational cost for the decomposition, we examined the actual decomposition time of randomly generated "fully connected graphs of diverse sizes" and presented their results in the pdf of the general response (Fig 1(b)). The decomposition was performed using Pytorch (with torch.linalg library) on one Nvidia T4 GPU. As shown in the result, a random graph with 10 nodes requires 0.0004s and a random graph with 5000 nodes requires 20.1s for decomposition. If there are 1000 graphs with 5000 nodes, the required time is probably <6 hours. We think this computational cost with ~5000 nodes may be reasonable and applicable in many practices as it only needs to be done only once at the first preprocessing stage.
>
> [1] Xu et al. "Graph wavelet neural network." International Conference on Learning Representations (ICLR), 2019.
>
> [2] Ma et al. "Learning multi-resolution graph edge embedding for discovering brain network dysfunction in neurological disorders." Information Processing in Medical Imaging (IPMI), 2021.
>
> Q2) How do you define $s$?
>
> A) The scales for graph wavelet bases were randomly initialized within a range $[10, 50]$, and they converged to different values after training. We provided figures of convergences of the scales for each dataset in Section 7 of the supplementary material. Also, the optimal values are presented in the pdf of the general response, which we will add in the revision once the paper is accepted.

---

> > ### Comment · Reviewer_6cNy · 2023-08-17
> > **Thanks for the authors' response.**
> >
> > Thank all authors for clarifying my concerns. I'd like to keep my score.

---

### Author Rebuttal · Authors · 2023-08-09

We thank all reviewers for their constructive reviews with anonymously positive evaluations.

In the pdf of the general response, we present $\bf{1) }$ A revised version of Fig. 2a and its description, $\bf{2) }$ an analysis on the computational time of eigendecomposition, $\bf{3) }$ optimal scale values after training, $\bf{4) }$ comparison with additional baselines, and $\bf{5) }$ comparison with an additional dataset.

---

### Decision · Program_Chairs · 2023-09-21

**Decision:**

Accept (poster)

**Comment:**

The paper proposes a new method for diffusion-based graph generation, by using graph wavelet bases to separate predictions of different frequency ranges, thereby overcoming some of the oversmoothing issues of existing methods. Reviewers were generally aligned and found the paper interesting and novel.